# Body Fat Percentage and Availability of Oral Food Intake: Prognostic Factors and Implications for Nutrition in Amyotrophic Lateral Sclerosis

**DOI:** 10.3390/nu13113704

**Published:** 2021-10-21

**Authors:** Jin-Woo Park, Minseok Kim, Seol-Hee Baek, Joo Hye Sung, Jae-Guk Yu, Byung-Jo Kim

**Affiliations:** 1Department of Neurology, Korea University Anam Hospital, Korea University Medicine, Seoul 02841, Korea; parkzinu@korea.ac.kr (J.-W.P.); virgo0906@korea.ac.kr (S.-H.B.); centertruth@korea.ac.kr (J.H.S.); 2Rodem Hospital, Incheon 22142, Korea; k.minseok@wustl.edu; 3BK21 FOUR Program in Learning Health Systems, Korea University, Seoul 02841, Korea

**Keywords:** amyotrophic lateral sclerosis, body fat percentage, oral food intake, survival

## Abstract

Adequate nutritional support and high body mass index (BMI) are good prognostic factors for disease progression and survival in amyotrophic lateral sclerosis (ALS). However, whether the composition of body weight, such as body fat percentage, has an independent effect on ALS prognosis remains unclear. The clinical data of 53 ALS patients were collected by medical record review. The data included: disease onset, sex, age, time of diagnosis, survival duration, presence of percutaneous endoscopic gastrostomy (PEG), nasogastric tube, tracheostomy, and availability of oral intake throughout the course of the disease, and interval measurement values of body mass by bioelectrical impedance analysis (BIA). The interval change (∆) of the BIA parameters was calculated by subtracting the follow-up values from the baseline values. Change in body fat percentage/interval between BIA measurements (months) (hazard ratio [HR] = 0.374, *p* = 0.0247), and availability of oral food intake (HR = 0.167, *p* = 0.02), were statistically significant for survival duration in multivariate hazard proportional regression analysis. Survival analysis and Kaplan–Meier curves showed similar results. Higher average monthly change in body fat percentage and availability of oral food intake are prognostic factors in ALS survival.

## 1. Introduction

ALS is a neurodegenerative disorder characterized by progressive motor neuron degeneration, resulting in progressive weakness with muscle mass loss and eventual death [1,2,3,4]. However, the etiology and prognostic factors of ALS are poorly understood, and the treatment options are limited [5]. 

Previous studies have suggested that complete general care with adequate nutritional support could delay muscle mass loss, and increasing body weight or body mass index (BMI) may improve the prognosis for ALS patients [6,7,8,9]. However, these studies did not analyze the effect of each component of body weight (e.g., muscle mass, water, fat, etc.) on ALS prognosis. 

There is a close correlation between BMI and body weight; however, an increase in BMI or body weight does not directly indicate an increase in specific body composition, such as body fat mass [10]. Measuring only BMI as a marker for obesity is considered insufficient since an increase in muscle mass or water content can increase BMI. Additionally, a paradox of low BMI and high body fat percentage has been suggested [11], indicating that BMI is not always a good surrogate marker for obesity. However, most studies on ALS have generally accepted that increased BMI is a marker of overweight status and obesity, and an indicator of good nutritional status, regardless of the effect of each component of body weight [12,13].

Therefore, this study aimed to investigate whether changes in body composition are associated with the prognosis of ALS patients.

## 2. Materials and Methods

Among 220 ALS patients who were admitted to a single community hospital (Rodem Convalescent Hospital, Incheon, Korea) for general care between 2013 and 2021, patients who met the inclusion and exclusion criteria for this study were selected from the retrospective data review. All patients met the diagnostic criteria of the revised El-Escorial criteria for clinically possible ALS or more [14]. Patients who underwent initial body bioelectrical impedance analysis (BIA) within one month after hospital admission were included in the study. BMI was measured at the time of the initial BIA measurement. Only patients who were given a sufficient caloric diet were included in the study to eliminate the potential influence on the results of individual feeding habits or malnutrition. In brief, patients whose BMI was <23 kg/m^2^ at the time of initial BIA measurement were fed a sufficient caloric diet (1800–2000 kcal per day) targeted to a BMI of 25 kg/m^2^. Patients who could not perform the recommended rehabilitation exercise programs were excluded to avoid muscle wasting due to reduced activity or being bedridden. Other exclusion criteria included any medical condition that may affect body weight, such as a history of psychiatric illness, autoimmune disease, severe diabetes mellitus, thyroid dysfunction, and active infections at the time of hospital admission. This study was approved by the Institutional Review Board of the Public Institutional Bioethics Committee designated by the Ministry of Health and Welfare (P01-202106-21-010). Written informed consent was waived since this was a retrospective study.

BIA data were collected using two multifrequency 8-electrode BIA (InbodyS10^®^ system, InBody Corp, Seoul, Korea). The details of this method and its validity have been previously described in detail elsewhere [15,16]. The principle of BIA measurement assumes that the composition of the body tissue has a different amount of water content, thereby producing different levels of resistance and reactance to currents of different frequencies. A total of 30 impedance measurements were obtained at six different frequencies (1 kHz, 5 kHz, 50 kHz, 250 kHz, 500 kHz, and 1000 kHz) for five segments of the body (right and left arms, right and left legs, and trunk). The measurement was done two hours after a meal, and patients were instructed not to exercise excessively to minimize the effect of other factors that may influence the BIA values [16,17]. The parameters obtained in the measurement included the following: weight, lean muscle mass (muscle mass), skeletal muscle index (skeletal muscle mass/height [m^2^]) [18], fat-free mass, skeletal muscle mass, body fat percentage, extracellular water/total body water ratio (ECW/TBW), basal metabolic rate, and phase angle arc tangent of (Xc/R) × 180°/π), which were automatically calculated by the device. Individual participants’ weights were measured using a validated device (CI-2001AS/BS and STUB-II, CAS, Seoul, Korea). Tape measurement by the designated protocol was used to measure the height of patients who were unable to stand up [19]. BMI was calculated by dividing body weight by height. The first BIA measurement was assessed within one month after admission to the hospital. Follow-up BIA data were collected from the available BIA results at least one month after the initial data collection. 

Disease onset (i.e., symptom onset) was defined by the year and month of individual symptom onset related to ALS based on the patients’ memory. Time to event was calculated by measuring the duration from disease onset to death or the censor date (5 March 2021). Sex, age, symptom onset, time of diagnosis, and survival duration (i.e., time to event [months]) were also acquired. The interval change (∆) from BIA measurements was calculated by subtracting follow-up measurements from baseline values, including the following: ∆ BMI, ∆ skeletal muscle mass, ∆ fat-free muscle mass, ∆ phase angle, ∆ ECW/TBW, and ∆ basal metabolism were obtained. The presence of percutaneous endoscopic gastrostomy (PEG), nasogastric tube, tracheostomy, availability of oral intake throughout the course of the disease, and time interval (months) between the first and final BIA measurements were also collected for analysis.

Linear regression analysis was conducted to determine the association between interval duration (months) among the BIA measurements and change in BIA parameters (∆ skeletal muscle index, ∆ muscle mass, ∆ fat-free mass, ∆ skeletal muscle mass, ∆ body fat percentage, ∆ basal metabolism rate, ∆ phase angle, ∆ weight, ∆ BMI, ∆ ECW/TBW). To evaluate the factors affecting individual survival, the hazard ratio (HR) was assessed with 95% confidence intervals (CIs). Age, sex, average monthly changes in BIA parameters, presence of tracheostomy, and availability of oral food intake were used as variables, and survival duration (from symptom initiation to death) was the primary outcome measurement. Univariate and multivariate analyses were performed using Cox proportional hazard models. For the multivariate Cox proportional hazard model, only the statistically significant variables based on univariate analysis were included in the analysis. Kaplan–Meier curves for statistically significant variables remaining after multivariate Cox regression analysis were generated based on the multivariate Cox proportional hazard model results. The groups were divided into two subgroups based on the values of the variables above and below the median cutoff points and compared using the log-rank test and Gehan–Breslow–Wilcoxon test. Statistical significance was set at *p* < 0.05, and all statistical analyses were performed using SAS (ver. 9.4, SAS Institute Inc., Cary, NC, USA).

## 3. Results

Among 220 ALS patients who were admitted during the study period, 53 who satisfied the inclusion and exclusion criteria were included in the final analysis of the data (Figure 1). 

The mean age of the patients with ALS (*n* = 53) at symptom onset was 61.25 ± 10.24 years old. The mean duration between baseline and follow-up BIA measurements was 7.46 ± 5.08 months. The demographics of the patients’ information and BIA results are described in detail in Appendix A.

In the simple linear regression analysis, ∆ skeletal muscle index (*p* = 0.0012, R^2^ = 0.191), ∆ muscle mass (*p* = 0.0027, R^2^ = 0.1661), ∆ fat-free muscle mass (*p* = 0.0036, R^2^ = 0.1576), ∆ skeletal muscle mass (*p* = 0.0012, R^2^ = 0.192), ∆ body fat percentage (*p* = 0.0147, R^2^ = 0.1134), ∆ basal metabolism rate (*p* = 0.0036, R^2^ = 0.1571), and ∆ phase angle (*p* = 0.0308, R^2^ = 0.09) showed a statistically significant negative correlation with the time interval between the BIA measurements, while change in body fat percentage showed a positive correlation (*p* = 0.0147, R^2^ = 0.1134), indicating that there is a tendency for time-related changes in these BIA parameters (Figure 2). In other words, skeletal muscle index, muscle mass, fat free mass, skeletal muscle mass, basal metabolism rate, and phase angle tended to decrease in a time-dependent manner.

However, no close relationship was observed between ∆ weight (*p* = 0.7704, R^2^ = 0.0017), ∆ BMI (*p* = 0.6712, R^2^ = 0.0036), and ∆ ECW/TBW (*p* = 0.3737, R^2^ = 0.0159), and the time interval of each measurement.

Univariate analysis of survival using the Cox proportional hazard model showed that the availability of oral food intake (HR = 0.228, *p* = 0.0461), average monthly change in body fat percentage (∆ body fat percentage/interval between BIA measurements [months], HR = 0.418, *p* = 0.023), average monthly phase angle change (∆ phase angle/interval between BIA measurements [months], HR = 0.429, *p* = 0.0374), and average monthly ECW/TBW change (∆ ECW/TBW/interval between BIA measurements [months], HR = 2.871, *p* = 0.012) were statistically significant predictors of longer survival in ALS patients. Among these variables, only the availability of oral food intake (HR = 0.098, *p* = 0.0073) and average monthly body fat percentage change (HR = 0.349, *p* = 0.0274) remained significant in the multivariate analysis (Table 1). 

The Kaplan–Meier analysis revealed that survival was significantly longer in ALS patients who were able to be fed orally (HR = 4.711, *p* for log-rank test = 0.03, *p* for Gehan–Breslow–Wilcoxon test = 0.0342) and who had a relatively higher average monthly change in body fat percentage change (HR = 4.898, *p* for log-rank test = 0.0269, *p* for Gehan–Breslow–Wilcoxon test = 0.0153), both in the log-rank test and Gehan–Breslow–Wilcoxon tests (Figure 3). 

## 4. Discussion

The main observations in this study were that a higher average monthly change in body fat percentage and availability of oral food intake were prognostic factors for longer survival in ALS patients admitted to a single convalescent hospital. In other words, the patients who did not at least lose their body fat percentage and who were able to eat orally survived longer in this study.

We also observed that the change in the parameters measured by BIA had a time-interval relationship, that is, the tendency of increase (e.g., ∆ body fat percentage) or decrease (e.g., ∆ skeletal muscle index, ∆ muscle mass, ∆ fat-free mass, ∆ skeletal muscle mass, ∆ basal metabolism rate, and ∆ phase angle) in BIA parameters closely correlated with the time interval between the measurements. 

Currently, there is no clear study that considers the average monthly change in body fat percentage as a prognostic factor linked to survival duration in ALS patients using BIA. Only one recent study demonstrated the potential of fat mass measurement with dual-energy X-ray absorptiometry (DXA) as a prognostic marker by showing that monthly change in amyotrophic lateral sclerosis functional rating scale-revised (ALSFRS-R) correlated significantly with monthly changes in the total fat mass index, percentage fat, and baseline fat percentage [20], which was consistent with our results. Our other major finding, the availability of oral food intake, is an essential prognostic factor that has already been generally accepted as a predictor of improved ALS clinical outcomes. 

As discussed previously, increased BMI may be inversely related to ALS risk and may be associated with a good prognosis in ALS [21]. In a recent meta-analysis, it was suggested that high BMI can significantly improve the long-term survival of ALS patients [22]. Rapid weight loss in ALS is also associated with poor prognosis and faster progression of the disease [23]. However, it is unclear whether changes in the specific composition of body weight are related to the prognosis of ALS [24]. Considering that body weight is composed of several components, this study may offer a clue for future research that increased body fat percentage, rather than increase of other components of body weight, is an important factor that may yield a good prognostic outcome in ALS. 

We did not aim to elucidate the pathomechanism; however, a priori, an assumption was made based on many in vitro and clinical studies regarding the possible neuroprotective effects of fat (i.e., adipose tissue). For example, several recent studies in this field have discussed leptin, a hormone produced by adipocytes, due to its possible neuroprotective role [25,26]. Leptin also exhibited neuroprotective effects on cerebral ischemia due to increased P-STAT3, PGC-1α, and matrix metalloproteases (MMPs), and decreased apoptosis in an in vivo study. Another recent study demonstrated the role of leptin in neuroprotection by repressing the integrin-linked kinase signaling pathway [27]. Previous studies have suggested that saturated fatty acids, monounsaturated fatty acids, and polyunsaturated fatty acids exhibit neuroprotective effects [28,29,30,31]. 

It should also be investigated whether improved prognosis in patients with a higher body fat percentage reflects better nutritional status or is an effect of an increase in body fat per se [32,33,34,35]. Many studies have shown that individuals with worse nutritional status may be at an increased risk of ALS [36,37]. However, it has also been reported that nutritional habits (e.g., preference for carbohydrates rather than fat) may be related to worse outcomes of ALS, while BMI alone may not [38,39]. Therefore, a more detailed approach and evaluation of the individual nutritional status are necessary. 

BMI is a good method for measuring individual body fat; however, due to low sensitivity in detecting body fat, DXA has been accepted as the gold standard for measuring body weight composition and individual nutritional status [40]. Several limitations have been reported when using DXA, including high equipment cost, radioactivity, and inability to perform repeated measurements within a short period [41]. On the other hand, the use of BIA has many benefits compared with DXA measurement in the study of ALS (e.g., feasibility, cheaper price, non-invasiveness). A recent study has also demonstrated a high correlation of BIA measurement results compared with those estimated by DXA, and the use of this method in ALS research has been validated [42,43]. 

The univariate analysis in this study showed that an increased average monthly change in body fat percentage and phase angle, availability of oral food intake, and decreased average monthly change in ECW/TBW were associated with longer survival duration in ALS. Although the average change in phase angle and monthly ECW/TBW did not remain significant after multivariate analysis, these results were interesting since they were consistent with those of previous studies. The phase angle has already been discussed as an independent poor prognostic factor of survival in ALS (i.e., was greatly decreased in ALS patients) [44]. Therefore, it can be assumed that an increased average monthly change in the phase angle is related to better prognosis in ALS. The ECW/TBW ratio in ALS has not been studied. However, a high ECW/TBW ratio is generally associated with poor prognosis in various disorders [45]. Therefore, it can be assumed that a nearly three-fold increase in HR in the univariate analysis has possible implications for the poor prognosis of ALS. 

The strength of our study is that all patients were from a single convalescent hospital specialized for ALS patients. Therefore, general lifestyles in the hospital are very similar among patients (e.g., diet and rehabilitation program). In addition, they used the same BIA device, which may reduce the bias between and within the participants. However, there are some limitations to this study owing to the retrospective study design. 

First, the interval between the BIA measurements was different for each participant. We used the average monthly change of the parameters rather than raw BIA data to decrease bias related to differences in measurement intervals among study subjects. Second, there was no control group in this study, and baseline measurements of the parameters were not collected immediately after symptom initiation or at the time of diagnosis. Instead, we considered BIA parameters as a baseline if they were analyzed within one month after admission to the hospital. Collection of detailed clinical data, such as amyotrophic lateral sclerosis functional rating scale-revised (ALSFRS-R), should be obtained in a future study to evaluate the slower progression of individual symptoms related to body fat percentage.

## 5. Conclusions

Our study results suggest that the average monthly change in body fat percentage and the availability of oral food intake are the most important prognostic factors in ALS survival. Sufficient oral food intake with adequate caloric support may be beneficial to ALS patients. Future clinical trials are needed to reveal the pathomechanism and confirm our results.

## Figures and Tables

**Figure 1 nutrients-13-03704-f001:**
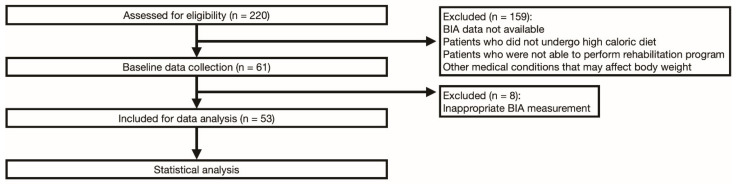
Consort flowchart of the study.

**Figure 2 nutrients-13-03704-f002:**
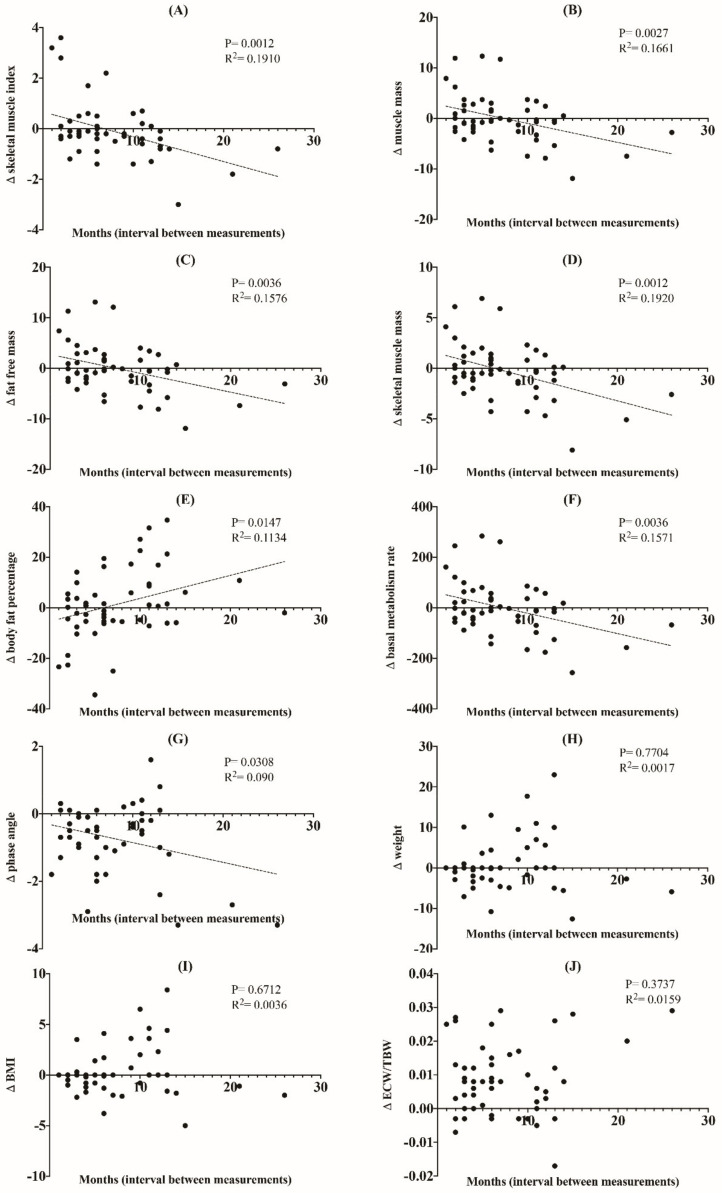
Linear regression showing changes in BIA parameters (skeletal muscle index (**A**), muscle mass (**B**), fat-free mass (**C**), skeletal muscle mass (**D**), body fat percentage (**E**), basal metabolism rate (**F**), phase angle (**G**), weight (**H**), BMI (**I**), ECW/TBW (**J**)) versus interval (months) between BIA measurements.

**Figure 3 nutrients-13-03704-f003:**
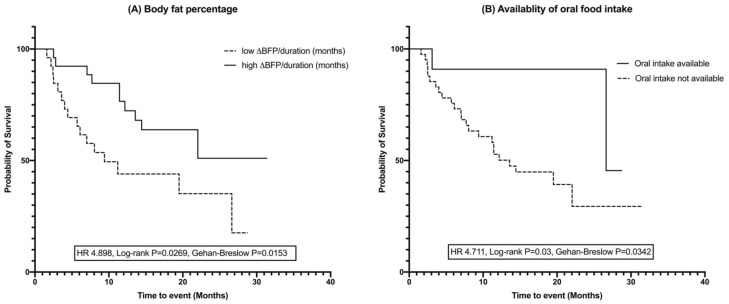
Kaplan–Meier curves and log-rank test for survival analysis showing the difference in the duration of survival in ALS patients: (**A**) Body fat percentage, (**B**) Availability of oral intake. Groups were divided based on the median values for body fat percentage. The higher body fat percentage and availability of the oral food intake increased the survival duration in ALS patients. ∆—interval change (follow-up – baseline values); BFP—body fat percentage.

**Table 1 nutrients-13-03704-t001:** Univariate and multivariate hazard proportional regression analyses of variables related to ALS survival.

Variables	HR	95% CI	*p*-Value
**Univariate Analysis**			
Age	1.047	0.997–1.099	NS
Gender	0.743	0.33–1.67	NS
∆ Body fat percentage/duration (months)	0.418	0.188–0.929	0.0323 *
Availability of oral food intake	0.228	0.053–0.975	0.0461 *
∆ Phase angle/duration (months)	0.429	0.194–0.952	0.0374 *
∆ ECW/TBW/duration (months)	2.871	1.261–6.535	0.012 *
∆ Skeletal muscle index/duration (months)	1.31	0.601–2.855	NS
∆ Muscle mass/duration (months)	1.763	0.808–3.846	NS
∆ Fat-free mass/duration (months)	1.448	0.667–3.146	NS
∆ Skeletal muscle mass/duration (months)	2.042	0.925–4.509	NS
∆ Basal metabolism rate/duration (months)	1.679	0.771–3.659	NS
∆ Weight/duration (months)	1.635	0.686–3.896	NS
∆ BMI/duration (months)	1.635	0.686–3.896	NS
Presence of tracheostomy	1.133	0.464–2.764	NS
**Multivariate Analysis**			
∆ Body fat percentage/duration (months)	0.374	0.158–0.882	0.0247 *
Availability of oral food intake	0.167	0.037–0.755	0.02 *
∆ Phase angle/duration (months)	1.215	0.177–8.344	NS
∆ ECW/TBW/duration (months)	3.03	0.429–21.384	NS

HR—hazard ratio; CI—confidence interval. HRs were adjusted for the listed variables. * *p* < 0.05.

## Data Availability

The data presented in this study are available upon request from the corresponding author. The data are not publicly available due to restrictions, such as privacy or ethics.

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
