# Peer review of "Body Fat Percentage and Availability of Oral Food Intake: Prognostic Factors and Implications for Nutrition in Amyotrophic Lateral Sclerosis"

_nutrients, 2021, doi:10.3390/nu13113704_

Round 1

Reviewer 1 Report

The manuscript is well designed and the data is very interesting. The results suggest that the average monthly change in body fat percentage
and the availability of oral food intake are the most important prognostic factors in ALS survival. Clinical data will undoubtedly be necessary for a future study

Author Response

We appreciate the reviewer’s compliments and appraisal of our work.

Reviewer 2 Report

Dear Authors

the presentation of the work does not appear so simple to read. I think you should better present your results and clarify the clinical implications. Moreover I suggest to create a comparison between the nutritive modifications with clinical and prognostic index of ALS. I suggest to investigate if there is modifications of laboratory biomarkers as albumin. Is there an increase of CV risk in these patients? 

Please write stronger discussion and conclusions suggesting a clinical impact of your work.

Author Response

We appreciate your valuable comments.

Q: The presentation of the work does not appear so simple to read. I think you should better present your results and clarify the clinical implications.

A: We added the paragraph in the results section to clarify our findings as follows: “In other words, skeletal muscle index, muscle mass, fat free mass, skeletal muscle mass, basal metabolism rate, and phase angle tended to decrease in a time-dependent manner (page 4, lines 131-132)”.

Q: I suggest to create a comparison between the nutritive modifications with clinical and prognostic index of ALS.

A: Unfortunately, as was described, the limitation of this study was the lack of a control group. Not only because this study was retrospectively designed, but also the study design with a nutritive modification is probably ethically difficult to perform.

Q: I suggest investigating if there is modifications of laboratory biomarkers as albumin.

A: As you have mentioned, serum plasma levels is one of an important indicator of nutritional status and it has been found that the levels of albumin are low in ALS patients compared to control. Although it was not presented in the manuscript, among the subjects, 12 patients were measured serial albumin levels (baseline and 12 months f/u). Mean albumin levels were 3.87 g/dL for baseline (at the time of admission) and 3.7 g/dL for f/u, both of which were within the normal range (3.5 – 5.2 g/dL) but were slightly low. The pairwise t-test revealed that there is no statistically significant difference between the measurements. Previous cohort study suggested that the albumin levels tend to decrease as a disease process and a decrease in albumin levels were associated with a higher risk of mortality. Considering this, the relatively spared albumin levels after f/u might have been influenced by sufficient nutritional support. However, we did not discuss this in our manuscript due to small numbers (n=12) for statistical analysis. Also, we assume this comparison is not sufficient to discuss the correlation of albumin and nutritional support since no control was available.

Q: Is there an increase of CV risk in these patients? 

A: We did not investigate the CV risk in this study. One could be concerned if the CV risk can be increased with the high caloric diet. To control those kinds of risk, we targeted not to exceed BMI more than 25 kg/m2 strictly as described in the methods section. The mean age at the initial body impedance analyzer measurement was 65 years old in this study. Considering that 2000 for men and 1600 for women is generally accepted as adequate calories per day for Koreans, the total calories of 1800-2000 per day is only slightly higher compared to those generally accepted. But we agree that the term ‘high caloric diet’ might confuse the readers. Therefore, we changed ‘high caloric diet’ to ‘sufficient caloric diet’ throughout the manuscript.

Q: Please write stronger discussion and conclusions suggesting a clinical impact of your work.

A: We added or deleted some paragraphs to make discussion and conclusion as follows:

Added:

“In other words, the patients who did not at least lose their body fat percentage and who were able to eat orally survived longer in this study (page 5, lines 166-167).”

“Sufficient food intake with adequate caloric support may be beneficial to ALS patients (page 7, lines 249-250).”

Round 2

Reviewer 2 Report

Dear Authors

Thank you for your activity of revision. I think the paper still has some gaps, in terms of clinical relapse and effectiveness.

Best regards

Author Response

Dear Reviewer 2,

Thank you very much for your comments for elaborating on this research.

Since the basic characteristic of amyotrophic lateral sclerosis is neurodegenerative (i.e., generally progressing disease without relapsing or remitting), it is inappropriate to discuss the relapse.

Please note that this is not an interventional study to find out the effectiveness of certain kinds of treatments or managements. Please also understand that this study was retrospectively designed and the main purpose was to investigate whether the changes in body composition are associated with the prognosis of ALS patients. Therefore, we assumed that to describe the clinical relapse and effectiveness in the manuscript is somewhat out-focused in these perspectives.